# The Role of P2X7 Purinoceptors in the Pathogenesis and Treatment of Muscular Dystrophies

**DOI:** 10.3390/ijms24119434

**Published:** 2023-05-29

**Authors:** Krzysztof Zabłocki, Dariusz C. Górecki

**Affiliations:** 1Laboratory of Cellular Metabolism, Nencki Institute of Experimental Biology, Polish Academy of Sciences, 02-093 Warsaw, Poland; k.zablocki@nencki.edu.pl; 2School of Pharmacy and Biomedical Sciences, University of Portsmouth, Portsmouth PO1 2DT, UK

**Keywords:** Duchenne muscular dystrophy, mdx, P2X7

## Abstract

Muscular dystrophies are inherited neuromuscular diseases, resulting in progressive disability and often affecting life expectancy. The most severe, common types are Duchenne muscular dystrophy (DMD) and Limb-girdle sarcoglycanopathy, which cause advancing muscle weakness and wasting. These diseases share a common pathomechanism where, due to the loss of the anchoring dystrophin (DMD, dystrophinopathy) or due to mutations in sarcoglycan-encoding genes (LGMDR3 to LGMDR6), the α-sarcoglycan ecto-ATPase activity is lost. This disturbs important purinergic signaling: An acute muscle injury causes the release of large quantities of ATP, which acts as a damage-associated molecular pattern (DAMP). DAMPs trigger inflammation that clears dead tissues and initiates regeneration that eventually restores normal muscle function. However, in DMD and LGMD, the loss of ecto-ATPase activity, that normally curtails this extracellular ATP (eATP)-evoked stimulation, causes exceedingly high eATP levels. Thus, in dystrophic muscles, the acute inflammation becomes chronic and damaging. The very high eATP over-activates P2X7 purinoceptors, not only maintaining the inflammation but also tuning the potentially compensatory P2X7 up-regulation in dystrophic muscle cells into a cell-damaging mechanism exacerbating the pathology. Thus, the P2X7 receptor in dystrophic muscles is a specific therapeutic target. Accordingly, the P2X7 blockade alleviated dystrophic damage in mouse models of dystrophinopathy and sarcoglycanopathy. Therefore, the existing P2X7 blockers should be considered for the treatment of these highly debilitating diseases. This review aims to present the current understanding of the eATP-P2X7 purinoceptor axis in the pathogenesis and treatment of muscular dystrophies.

## 1. Introduction

Muscular dystrophies (MDs) are a group of inherited neuromuscular disorders, presenting with a spectrum of symptoms. Although not all MDs are very severe diseases, the most common types cause progressive muscle weakness and wasting, leading to devastating disabilities that become life-threatening. In most cases, there is no cure for MDs, and the symptomatic treatment can be beneficial in some types only. The most common of MDs is Duchenne muscular dystrophy (DMD), which causes severe disability and premature death due to respiratory or cardiac failure. Additionally, DMD affects the nervous system, leading to neuropsychiatric abnormalities [1]. Duchenne MD is a consequence of out-of-frame mutations in the *DMD* gene located on the X chromosome. Therefore, it almost exclusively affects boys, with a prevalence of 1 per 5000 live births [2]. The *DMD* gene is composed of 79 exons and, in humans, contains eight promoters driving expressions of three full-length (427 kDa) dystrophins, a 412 kDa embryonic isoform [3], and four, progressively shorter, isoforms expressed in various tissues and playing specific roles. Loss of full-length dystrophins is both necessary and sufficient to cause DMD. Mutations affecting the shorter isoforms can exacerbate muscle pathology [4,5,6] and particularly cognitive and behavioral abnormalities [7]. In muscle, DMD mutations affect, to a different extent, all muscle cell types, including stem (satellite) cells, myoblasts [8], and myofibers. In myofibers, the 427 kDa dystrophin is a cytoskeletal protein, which tethers intracellular actin filaments with the dystrophin-associated protein (DAP) complex comprising of dystroglycans and sarcoglycans, located in the sarcoplasmic membrane. Further interactions involve specific intra- and extracellular proteins [9]. The lack of dystrophin prevents proper DAP formation, which is a scaffold and an important element of the cellular signaling network [1]. Through DAP, Duchenne shares molecular abnormalities with a subset of Limb-Girdle Muscular Dystrophies (LGMD). This is a group of MDs caused by mutations in the γ, α, β, and δ sarcoglycan genes, encoding the sub-complex of DAP proteins. These mutations result in sarcoglycanopathies LGMD-2C to LGMD-2F (LGMD R5, LGMD R3, LGMD R4, LGMD R6, respectively, according to the new nomenclature) [10,11]. The loss of one sarcoglycan protein triggers the disruption of the entire tetrameric sarcoglycan sub-complex. In the absence of dystrophin/DAPC, embryonic muscle development is affected [12] and these abnormalities are later recapitulated in muscle regeneration of adult dystrophic muscle. The lack of dystrophin initiates a pathological vicious cycle involving abnormalities in the asymmetric division of dystrophic satellite cells [13], which are impacting myoblasts [14], and differentiate myotubes, ultimately resulting in dysfunctional myofibers unable to resist the physiological contraction-induced stress [15,16].

The LGMD sarcoglycanopathies, albeit heterogeneous in the time of onset and in severity, also lead to significant muscle damage [10]. In both dystrophino- and sarcoglycanopathies muscle degeneration and regeneration occur in cycles, and these are further aggravated by significant sterile inflammation. The inflammatory cells are attracted by the danger/damage-associated molecular pattern (DAMP) molecules released from dead and damaged muscles. In any acute damage, skeletal muscle responds to injury with an inflammatory response. It is indispensable for muscle regeneration because it demarcates the injured area, removes dead cells, thus preparing tissue for repair, and finally specific inflammatory cytokines activate satellite (muscle stem) cells into proliferation to evoke tissue repair [16,17]. Various immune cells have roles in skeletal muscle regeneration, but macrophages are the most abundant and also the most significant population, involved across the entire process, from immediately after the initial injury until the late tissue re-modeling phase of muscle repair. These cells show remarkable phenotypic and functional plasticity throughout the entire process. This review aims to highlight the connection between the eATP-P2X7 purinoceptor axis in dystrophic muscle and inflammatory cells and its relevance to the disease pathomechanisms.

## 2. Sterile Inflammation Is a Hallmark of Dystrophic Muscles

Unfortunately, in MDs, the role of inflammation changes dramatically, not only because it becomes chronic in response to continuing muscle damage, but also because it can contribute to this damage [18]. Indeed, chronic inflammation is a hallmark feature in MDs. Dystrophic muscles of both patients and the mouse model of DMD (*Dmd^mdx^*) contain significantly more macrophages than muscles of the unaffected individuals [19]. In some sarcoglycanopathies, inflammatory changes can even exceed those found in dystrophin-deficient muscles [20]. Interestingly, in DMD, inflammatory changes precede the dystrophic muscle damage [21,22], which suggests that these may not be just the reaction to the dystrophic injury, but a representation of a specific dystrophic abnormality. There is some evidence that *Dmd^mdx^* macrophages exhibit trained immunity, which drives DMD inflammation [23]. Importantly, the sterile inflammatory response in MDs is not limited to the infiltrating immune cells but is also found in dystrophic muscles. In both DMD patients and in *Dmd^mdx^* primary muscle cells, inflammasome components were found up-regulated and the complex was overactivated [24]. Inflammasome is a general term describing a large multimeric protein complex that plays a key role in regulating the innate immune system and inflammatory signaling. Several inflammasome complexes have been described, of which the NLRP3 (nucleotide-binding domain leucine-rich repeat (NLR) and pyrin domain containing receptor 3) is relevant for MDs. The NLRP3 inflammasome complex consists of the NLRP3 protein sensor, the ASC adaptor (apoptosis-associated speck-like protein), and the caspase-1 effector. Formation of the NLRP3 inflammasome occurs in two stages: priming and activation. See recent comprehensive reviews for details [25,26]. Upon activation, NLRP3 oligomerizes and activates caspase-1, which initiates the proteolytic processing and release of pro-inflammatory cytokines IL-1β and IL-18 and can induce pyroptosis, an inflammatory form of cell death. Thus, the NLRP3 inflammasome activation plays a beneficial role in infectious diseases and some cancers, but it can be detrimental in the pathogenesis of inflammatory and metabolic diseases [26]. Indeed, aberrant NLRP3 activation can drive chronic inflammation, contributing to the pathology of inflammation-associated diseases, such as MDs [27]. Given that not only immune but also muscle cells can participate in inflammasome formation [24] and can release IL-1β, targeted treatments inhibiting NLRP-3 inflammasome using adiponectin [24] and MCC950 small-molecule inhibitor [28]. ameliorated symptoms in *Dmd^mdx^* mice. In addition to DMD, patients with dysferlinopathy, caused by mutations in the DYSF gene, also present with muscle inflammation and abnormal macrophage activation. Likewise, the dysferlin-deficient muscles show a significant up-regulation of inflammasome pathways [29]. The NLRP3 inflammasome overactivation, in both muscle cells and in infiltrating macrophages, was also found involved in the pathology of the Valosin-containing protein myopathy [30]. However, it must be noted that the disruption of ASC, the key inflammasome adaptor, showed minimal impact on muscle pathology in the *Dmd^mdx^* model [31]. The nature of this discrepancy is unclear, but it might involve the still-unsolved non-canonical NLRP3 role [32], which may not require ASC.

Yet, dystrophic inflammation, albeit damaging, cannot be blocked completely [33,34]. Treatments reducing inflammation reduced the progression of MDs [33,34,35,36,37,38,39] but its ablation exacerbated the phenotype [33]. This agrees with the aforementioned importance of specific inflammatory cells and their cytokines for muscle stem-cell activation and ultimately muscle regeneration. Thus, more targeted approaches exploiting the specific features of MD sterile inflammation could suppress the damaging effect, while enhancing its pro-regenerative potential. Importantly in this respect, the activation of NLRP3 inflammasome emerged as one of the novel therapeutic targets in MDs. The NLRP3 can be activated in response to diverse triggers, including DAMPs. Among those, extracellular ATP (eATP) is one of the most potent activators, which interacts with the purinergic receptor type 2, family X, subunit 7 (P2X7) [40]. This P2X7 purinoceptor, expressed by all muscle-infiltrating immune cells, was also found overexpressed on dystrophic muscle cells [41,42,43]. The blockade of this receptor alleviated the damaging inflammation and promoted muscle repair in a mechanism, which is discussed below.

## 3. Purinergic Signaling and Its Alterations in Dystrophic Muscles

The intracellular concentration of ATP within skeletal muscle (SM) is high (5–10 mM), because the main muscle function, contraction, requires energy that is delivered in the form of ATP. Under physiological conditions, the bulk of ATP is intracellular, with only small amounts being released [44]. Such extracellular ATP becomes a signaling molecule stimulating specific purine-responsive receptors. These purinergic receptors can be subdivided into the ionotropic (P2X) and the metabotropic (P2Y) subtypes. Activation of purinoceptors results in increased intracellular Ca^2+^, either the extracellular entering through ion channels (P2X) or that released from the intracellular stores (P2Y) (reviewed in Burnstock 2020) [44], which are then replenished by store-operated calcium entry (SOCE). Purinoceptor activation in a healthy muscle triggers a range of downstream effects. For example, purinergic receptor-mediated signaling influences the proliferation and differentiation of muscle cells in development [45] and post-natal, with specific receptors found in freshly isolated embryonic skeletal muscle cells [46] and cultured muscle [47] from various mammalian species. In mouse myotubes in vitro, an increase in intracellular Ca^2+^ concentration required for differentiation was linked to the activity of both ionotropic and metabotropic receptors [48], of which, respectively P2X4, P2X5, P2X7, P2Y1, and P2Y4 purinoreceptors were identified [49]. In differentiated muscles, specific purinergic receptors are involved in the formation and function of neuromuscular junctions (NMJ) [50,51] and ATP is also co-released with acetylcholine at NMJ, where it acts as a co-transmitter [52].

In the heart, the eATP-purinoceptor axis also plays a role. In rat embryonic cardiomyocytes, ATP responses were found to differ with developmental stages, which was linked to the differential expression of P2X2 and P2X4 proteins at the earliest stages and P2X1, P2X2, P2X4, P2X5, and P2X7 in cell isolated at embryonic day 18. In addition to these ion channels, P2Y1, P2Y2, P2Y4, and P2Y6 metabotropic receptors were found expressed and functioning in cardiomyocytes across developmental stages analyzed [53]. Subsequent studies revealed that purinergic receptor subtype expression can vary in cells from different heart regions and mediate several important effects, including negative and positive inotropic and chronotropic effects, influencing glucose transport and preventing hypertrophy. Notably, some of these effects are species-specific (Reviewed in [54]). However, these distinct and subtle physiological functions can be overshadowed by muscle pathology. Any muscle damage results in millimolar concentrations of eATP being released, and then its function changes, as it becomes an immune-activating signal. As mentioned, high eATP is one of the key DAMP molecules, specifically activating the P2X7 purinoceptor, which fully responds to high eATP concentrations only. Therefore, P2X7 is considered a danger/damage receptor. It is expressed by all immune cell types and is responsible for triggering the inflammatory response either in response to infection or as a sterile inflammation across a spectrum of pathologies [40,55,56]. Its activation not only stimulates the NLRP3 inflammasome pathway, as explained above, but upon prolonged activation by high eATP, P2X7 can form a large pore (LP) and trigger plasma membrane permeabilization [57]. Therefore, to prevent such catastrophic events, the eATP is rapidly eliminated by extracellular hydrolyzing enzymes—nucleotidases (Reviewed in [58,59,60]). In healthy muscle, the main membrane-bound ATP hydrolase is α-sarcoglycan, which degrades around 25% of eATP found in muscle [61,62]. As a DAP protein, α-sarcoglycan is lost from the sarcolemma in both DMD and LGMDs. Thus, the imbalance between ATP release and degradation in these MDs is swayed dramatically (Figure 1). The resulting high eATP levels can over-activate purinoceptors present in muscle cells and contribute to the abnormal intracellular Ca^2+^ homeostasis found in dystrophic muscles [63].

## 4. Altered P2X7 Expression and Function in Dystrophic Cells

Importantly, in addition to the lost ecto-ATPase activity of α-sarcoglycan, a purinergic receptor abnormality was identified in a range of dystrophic cells. Several studies using a combination of molecular, biochemical, functional, and immunodetection methods identified alterations in the expression and function of specific purinoceptors in dystrophic muscle [64,65,66]. In contrast to their aforementioned physiological expression, ionotropic P2X2, P2X5, and P2X6 [45] as well as metabotropic P2Y2 and P2Y4 receptors [65], were found altered in dystrophy. It was found that immortalized mdx myoblasts exhibit substantially higher P2Y2 expression and activity than the control cells, and its activity is involved in regulating myoblast migration [67]. Similar effects were found in primary myoblasts isolated from specific mouse hind leg muscles: tibialis anterior, soleus, gastrocnemius, and flexor digitorum brevis. However, the distribution pattern and purinoceptors activity varied between different muscles [68]. More recently, the α-SG and γ-SG loss-of-function mutations were also shown to be associated with P2Y2 overexpression in mouse and human primary and immortalized myoblasts, respectively [69]. Yet, the most striking purinergic alteration was the up-regulated expression of the P2X7 receptor. It was noted in muscle biopsy samples from DMD patients [42,43], in mouse models of dystrophinopathy [41,42], and in α-sarcoglycanopathy [70]. In DMD, in addition to muscle cells, significant functional receptor abnormalities were found in lymphoblasts [71]. This might indicate that the loss of dystrophin expression is a causative agent for the purinergic abnormality presenting across species and cell types.

Dystrophic *Dmd^mdx^* myoblasts exposed to high eATP levels responded with increased cytosolic Ca^2+^ influx [72]. Treatment of *Dmd^mdx^* cells with an eATP degrading enzyme apyrase reduced the increases in intracellular Ca^2+^ levels [71] establishing the role of eATP. The P2X7 involvement was confirmed by the same response being evoked when using BzATP, an agonist preferentially activating P2X7 and conversely, being blocked by P2X7-specific inhibitors. The LP opening sensitive to pharmacological blockers, further confirmed P2X7 involvement [42]. P2X7 activation in dystrophic muscle cells under specific conditions led to cell death via necrosis or triggered a unique mechanism of autophagic cell death [72]. Interestingly, exposure of *Dmd^mdx^* myoblasts to high eATP concentrations caused rapid cell detachment, while lower concentrations alter the migration of dystrophic cells [manuscript in preparation]. Thus, large amounts of eATP continuously released from damaged dystrophic muscle cells, combined with reduced degradation of eATP and the significant P2X7 overexpression in dystrophic cells result in the overactivation of this receptor. Such a prolonged activation of P2X7, with its LP opening, can further increase the permeability of the dystrophic sarcolemma, contributing to the death of dystrophic myofibers, while death and abnormal migration of dystrophic myoblasts can further reduce the repair capacity and therefore contribute to inefficient regeneration of dystrophic muscles.

In addition to this direct muscle impact, continuously high eATP stimulates P2X7 purinoceptors in the immune cells, triggering chronic inflammatory responses. This, as explained earlier, exacerbates disease symptoms. Given that not only immune but also muscle cells participate in NLRP3 inflammasome formation [24], the resulting pyroptosis contributes significantly to dystrophic muscle-wasting [73]. In addition to pyroptosis, necroptosis has been identified as an important cell death mechanism in DMD [74]. The latter is activated by the sustained elevation of intracellular Ca^2+^, and therefore can be a direct result of calcium overload found in dystrophin-deficient cells [63]. Unsurprisingly, P2X7 purinoceptor alterations were found involved in the pathogenesis of DMD [75], LGMD [76], but also dysferlinopathy [29]. Although the pathological involvement of P2X7 receptors expressed on the inflammatory cells is easily explained, the function of P2X7 up-regulation in dystrophic muscle cells is puzzling.

## 5. P2X7 Up-Regulation: A Dystrophic Abnormality or Compensatory Adaptation?

P2X7 up-regulation in dystrophic muscle might be an effect of the loss of dystrophin and/or DAPC scaffolding altering the P2X7 membrane localization that triggers its overexpression and affects its function. However, there is no evidence of direct or indirect interaction between P2X7 and dystrophin and/or DAPC members. Second, enhanced P2X7 expression and signaling also occur in myoblasts [41,42] and lymphoblasts [71], cells that do not have detectable dystrophin/DAPC scaffolds, and in fact, abnormalities in myoblasts are likely to be epigenetic [14]. Furthermore, the fact that dystrophic purinergic alterations involve various receptors and, in a muscle-specific fashion [68], does not fit with the absence of dystrophin across all the muscles. All these findings suggest a different explanation. It might be that the P2X7 purinoceptor is up-regulated not as a result of the loss of dystrophin/DAP interactions but as a compensatory mechanism, an adaptation to highly unfavorable conditions, which are present in dystrophic muscles, and which include reactive oxygen species, inflammation, and metabolic abnormalities.

Analysis of the consequences of P2X7 overactivation in *Dmd^mdx^* myoblasts and myotubes identified eATP-induced P2X7-dependent autophagic flux, leading to CASP3-CASP7-independent dystrophic cell death. This P2X7-evoked autophagy was triggered by LP formation in a heat-shock protein HSP90-dependent fashion and was specific to dystrophic muscle cells [72]. Autophagy is a mechanism by which cellular components, organelles, but also debris are sequestered within autophagosomes, targeted for lysosomal degradation, and subsequently reused. Muscle injury triggers autophagy and its induction is necessary for successful muscle fiber recovery [77]. Interestingly, impaired autophagy has been identified in both dystrophic myoblasts [78] and myofibers, apparently being triggered by oxidative stress [79]. Then, there is evidence that proteins and organelles accumulating after muscle damage due to deficient autophagosome clearance reduce the regenerative capacity. Simply, the resulting shortage of energetic substrates and/or signaling molecules reduces satellite cell activation and differentiation [77]. Therefore, the following scenario could be envisaged: P2X7 overexpression in damaged muscles is a compensatory mechanism leading to eATP-induced P2X7-dependent enhanced autophagic flux. P2X7 activation by a brief exposure to eATP released as a result of transient and mild muscle damage would be fully controlled by ecto-ATPases, and thus result in a self-limiting stimulation of autophagy. The enhanced recirculation of damaged organelles would aid muscle growth and regeneration.

In contrast, in a chronic damage state caused by dystrophino-, sarcoglycano-, and dysferlinopathy, excessive amounts of eATP due to an increased release and reduced degradation combined with P2X7 up-regulation lead to prolonged, chronic stimulation and overactivation of autophagy, inducing the dystrophy-specific autophagic muscle death pathway (Figure 2).

In addition, in the heart, eATP-evoked P2X7 receptors activation prior to ischemia-reperfusion injury (preconditioning) was shown to protect cardiac muscle. Such preconditioning induces the release of endogenous cardioprotectants from cardiomyocytes in a process that involves P2X7 receptor signaling and coupling with pannexin-1 hemichannel opening [82,83]. The P2X7 up-regulation may suggest a similar protective mechanism being active in MDs. Interestingly, the seemingly paradoxical overexpression of P2X7 in dystrophic muscle aligns with its well-known overexpression on cancer cells. In this case, the overexpression of P2X7 also coincides with elevated eATP levels, yet does not trigger the P2X7-evoked cancer cell death cascade either. Different tumors seem able to take advantage of the P2X7 stimulation that provides significant benefits including increases in growth, migration, and invasion capability and, importantly, the Warburg effect (reviewed in [84]). This altered metabolic pathway involves energy production through a process of aerobic glycolysis consisting of increased glucose uptake and fermentation of glucose to lactate observed even in the presence of fully functioning mitochondria.

Yet another mechanism might play a role. Stimulation of P2X7 was found to improve myofiber metabolism and enhance satellite cell proliferation and differentiation in SOD1 mice. The metabolic mechanism in DMD may be analogous, whereby P2X7 signaling inhibits glycogen synthesis in favor of glucose consumption and improves muscle fiber mitochondrial respiration [85]. Indeed, DMD impacts muscle cell energetics significantly [86,87]. Given that there are questions about whether these mitochondrial alterations in DMD are a causative metabolic defect or an adaptive reprogramming process (Reviewed in [88]) P2X7 modulation might be one of the compensatory mechanisms alleviating dystrophic abnormalities.

Another indirect indication that P2X7 purinoceptors may play a protective role is that this receptor activity was found important for preventing ectopic calcification, which is one of the less-known features of DMD muscle pathology. Interestingly, in this case, it was P2X7 present in infiltrating inflammatory cells rather than in dystrophic muscle cells [6,89]. It is also worth noting that hyperactivation of P2X7 may not result in catastrophic LP opening and the death of dystrophic myofibers. It was found that prolonged activation of P2X7 can increase both the expression and release of active matrix metalloprotease 2 (MMP-2) [90]. The MMP-2 was shown to functionally inhibit this purinoceptor by cleaving it, (Figure 3).

Given that it was found in dystrophic muscle cells but also in macrophages, cancer, and even in HEK cells expressing this receptor artificially, the MMP2-evoked P2X7 degradation seems to be an important regulatory mechanism [89]. Moreover, the serum contains high levels of active MMP2. Therefore, it is possible that not all the mechanisms associated with the high-level P2X7 stimulation in dystrophic cells in vitro are active in the same way in dystrophic muscles or lymphoblasts in vivo, where P2X7 can be degraded by serum-born MMP2. Further studies are needed to follow pathways contributing to the dysregulated homeostasis in dystrophic muscles, as these could become specific therapeutic targets. However, irrespective of whether the effects of P2X7 activation established in vitro are fully replicated in vivo, the beneficial effects of P2X7 receptor blockade across several disease mechanisms have been proven in several mouse models of MDs in vivo.

## 6. Therapeutic Effects of P2X7 Blockade in Dystrophino- and Sarcoglycanopathies

This therapeutic impact was demonstrated in mouse models of dystrophino- and sarcoglycanopathy using genetic ablation and pharmacological blockade of P2X7 receptors. Double negative (*Dmd^mdx^*/*P2rx7^−/−^*) mice presented with a significant attenuation of dystrophic symptoms, which included reduced muscle damage, improved regeneration capacity, reduced creatine kinase serum levels (a marker of sarcolemma damage) as well as reduced inflammatory and pro-fibrotic molecular signatures. There was an overall decrease in the numbers of infiltrating macrophages in muscles from *Dmd^mdx^*/*P2rx7^−/−^* mice compared to *Dmd^mdx^* controls but also a shift in immune cell populations. First, the ratio of pro-inflammatory to pro-regenerative macrophages was significantly lower in *Dmd^mdx^*/*P2rx7^−/−^* muscles. Moreover, there was a shift from T cytotoxic towards T regulatory (T_reg_) cells, which was represented by significantly increased Foxp3 and IL-12α expression in dystrophic muscles with the receptor ablated [75]. Thus, P2X7 ablation not only ameliorated the damaging tissue inflammation but also promoted T_reg_ cell expansion, which has been independently shown to suppress dystrophic muscle damage [91].

Importantly, all improvements were evident both at the peak of disease severity and in old animals, in the leg, diaphragm, and cardiac muscles. This late effect is of particular importance as longer-surviving patients often develop and eventually succumb to cardiac failure. Notably, histologically identified reduction in the inflammatory markers and the improved muscle structure corresponded with significantly increased muscle strength in vivo [75]. Furthermore, *Dmd^mdx^*/*P2rx7^−/−^* mice showed improvements in specific cognitive and behavioral tests. This represents a unique example of a treatment that corrects both the muscle and non-muscle dystrophic phenotype, with severe CNS abnormalities, when present, with a profound and appalling impact on patients’ and their families’ quality of life (see below).

Furthermore, pharmacological blockade using P2X7 antagonists such as Coomassie Brilliant Blue G (BBG) and oxidized ATP (oATP) [43] produced improvements, even after a short-term treatment of *Dmd^mdx^* mice. This effect of the pharmacological treatment also included reduced tissue inflammation and an increased number of T_reg_ cells [43], corresponding with the genetic ablation data. Likewise, treatment with a broad-spectrum P2X receptors antagonist of the α-sarcoglycan knockout mouse model of LGMD, reduced muscle inflammation and promoted T_reg_ expansion. Again, this specific anti-inflammatory response with alteration of the adaptive immune component of muscle-infiltrating cells reduced necrosis and fibrosis and increased muscle strength in vivo [70]. Given that a broad-spectrum P2X blocker (oATP) was used in this study, these effects could be attributed to the blockade of the P2X4 purinoceptor present in inflammatory cells [92]. However, significant improvements were subsequently reported in the same model using the highly selective P2X7 purinoreceptor antagonist A438079, confirming the key role played by this receptor. After long-term treatment, muscle strength recovered to almost wild-type levels [76].

Taken together, these data provide a strong impetus to exploit P2X7 purinoceptor blockade as an attractive therapeutic target in dystrophino- and sarcoglycanopathies. Importantly, it is also one with great translational potential. First, several selective P2X7 inhibitors have been developed (e.g., GSK1482160; CE-224,535, and AZD9056) and proven safe in clinical trials in inflammatory pain, rheumatoid arthritis, and Crohn’s disease [93,94,95]. Several more drugs are in development. However, while these compounds would offer the most specific treatment, none have been approved as a medicine yet. Moreover, none was tested in children and therefore would need lengthy clinical trials. Therefore, the identification of existing, ideally pediatric, medicines that can block P2X7 purinoceptors, and their careful repurposing may be a more clinically relevant solution. For example, the well-known nucleoside reverse transcriptase inhibitor azidothymidine (AZT, Zidovudine), binds to the same allosteric site [96] on the P2X7 molecule as the specifically developed compounds [97] and is an effective P2X7 blocker [98]. AZT has a well-established pharmacological profile, including in the pediatric population. Short-term treatment of *Dmd^mdx^* mice with AZT decreased inflammation in the leg and heart muscles and reduced sarcolemma permeability. Moreover, such a treatment increased muscle strength and did not cause any detectable side effects [97]. These highly promising results in the preclinical trial in dystrophy fit with the findings that AZT treatment reduced age-related macular degeneration via a P2X7-dependent mechanism [99]. It appears that this drug is a prime candidate for rapid repurposing as a clinically relevant treatment of diseases requiring P2X7 purinoceptor inhibition. AZT is also a very low-cost treatment. It is worth noting given that cost-effectiveness is an important consideration when adopting new treatments and that high costs of medicines prohibit their application in less-developed counties. P2X7 blockade might potentially have an additional, important application. The main, currently pursued treatment for DMD is the replacement of dystrophin in muscles using genetic approaches. However, dystrophin re-expression evoked by exon skipping or gene therapy occurs in genetically deficient dystrophic muscles. Although a healthy skeletal muscle is in an immunogenic location, this is further exacerbated in DMD muscle, with its chronic inflammation. This results in the immune response being triggered against the re-expressed dystrophin, which is recognized as a neoantigen. This has a negative impact on treatment efficacy [100] and is potentially dangerous. Therefore, immunosuppression of these responses is being considered. However, induction of dystrophin-specific tolerance would maximize the treatment impact while minimizing the well-known risks of immunosuppression. Given that P2X7 receptor blockade induced Treg cells in dystrophic muscles [43,75] and that Treg up-regulation is one of the established mechanisms evoking immune tolerance [101,102,103,104,105,106] P2X7 blockade during dystrophin re-expression could have a combined therapeutic effect. By directly reducing the dystrophic inflammation and muscle pathology, while improving transgene expression via Treg expansion, such treatment could be more effective.

## 7. P2X7 in Dystrophic Brains

The impact of P2X7 has also been noted in relation to neuropsychological impairment associated with DMD [6]. As mentioned, ablation of P2X7 receptors corrected the cognitive and some aspects of the behavioral impairment in *Dmd^mdx^*/*P2rx7^−/−^* mice [75]. Currently, this is the only clinically viable treatment for this debilitating DMD abnormality. However, subsequent studies illustrated the complex nature of P2X7 involvement in this still poorly understood dystrophic brain defect. In contrast to the dystrophic muscle, *P2rx7* gene expression was found unaltered in *Dmd^mdx^* samples and, in fact, it decreased in dystrophin-null brains [107]. Intriguingly, loss of all dystrophins is known to be associated with more severe CNS abnormalities compared to the classical DMD phenotype in patients and in *Dmd^mdx^*, which lack the full-length dystrophins only [5]. Therefore, P2X7 seems to have a very different role in the dystrophin-null brain than in muscles, where its expression was found up-regulated [89]. Although its expression in the brain might not be up-regulated, P2X7 function there seems important, given that ablation of this receptor resulted in improved cognition [69]. This improvement might be a function of the damaging impact of eATP and/or inflammatory mediators released from a damaged and inflamed muscle that crosses the blood–brain barrier, which is permeable in dystrophic brains [108]. Understanding this mechanism should help in developing effective treatments for this neuropsychological condition. It is essential given that severe impairment affects one third of patients, further reducing the quality of life of patients and their families, and it is not tackled by any of the currently developed DMD treatments.

## 8. Conclusions

The dystrophic muscle microenvironment features elevated levels of eATP leading to high P2X7 purinoceptor activity, triggering chronic inflammation, which exacerbates dystrophic muscle pathology. Studies in mouse models of dystrophino- and sarcoglycanopathies have demonstrated that P2X7 blockade reduces damaging inflammation while promoting the pro-regenerative arm of the inflammatory response. Concomitantly, it reduces damage to myofibers, and boosts the regenerative potential of dystrophic myoblasts. P2X7 inhibition also alleviates cognitive and behavioral impairments in DMD and even improves the dystrophic bone defect [75]. This therapeutic impact across MDs on both muscle and non-muscle abnormalities is exceptional, but these wide-ranging therapeutic effects simply reflect the involvement of P2X7 activation in various disease processes. Although the mechanisms continue to be studied, a therapeutic strategy that modifies both muscle and non-muscle symptoms and applies across DMD and LGMD-2C-F, the two most common and debilitating groups of muscular dystrophies, is highly significant clinically. Importantly, P2X7 therapy is not constrained by causative mutations in DMD and LGMDs, and therefore is suitable for all patients. Thus, P2X7 is an attractive therapeutic target with at least one established medicine (AZT) that is ready for repurposing.

## Figures and Tables

**Figure 1 ijms-24-09434-f001:**
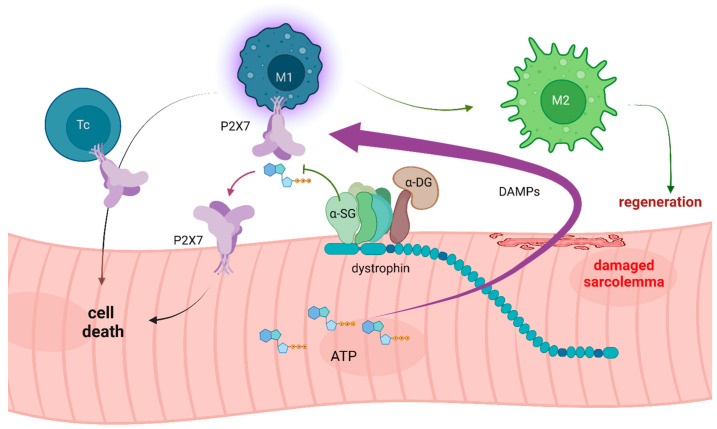
Loss of dystrophin and/or sarcoglycans (SG) destabilizes myofiber sarcolemma. DAMPs, including large quantities of ATP, are released through the damaged sarcolemma. Loss of ecto-ATPase activity of α-sarcoglycan occurring indirectly (dystrophinopathy) or directly (sarcoglycanopathy) reduces eATP hydrolysis. High eATP activates P2X7 receptors, and, combined with other DAMPs, triggers inflammation. The infiltrating macrophages (M1) and cytotoxic T-lymphocytes (Tc) contribute to muscle damage. Moreover, P2X7 up-regulated on dystrophic muscle cells, combined with high eATP levels, exacerbates the dystrophic muscle damage. However, inflammation cannot be eliminated, as the inflammatory response (M2 macrophage) is required to induce muscle regeneration. Figure generated in BioRender.

**Figure 2 ijms-24-09434-f002:**
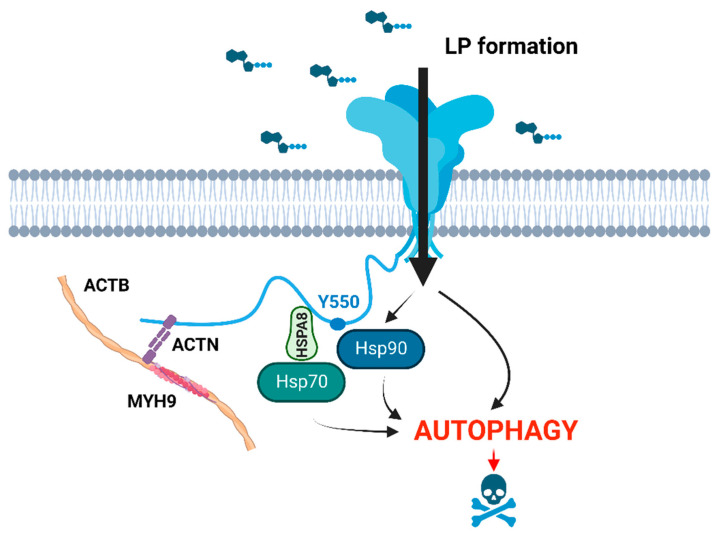
In dystrophic muscle cells, P2X7 activation by high eATP results in HSP90-mediated LP formation and autophagic flux, ultimately leading to cell death. The relevant known interactome at the C-terminus of P2RX7 involves ACTB, ACTN (actinin, α), MYH9 (myosin, heavy polypeptide 9, non-muscle), and heat-shock proteins HSP70 (HSPA2), HSPA8, and HSP90, where Y550 residue in the P2X7 C-terminus plays a role in mediating HSP90 phosphorylation status [80,81], BioRender.

**Figure 3 ijms-24-09434-f003:**
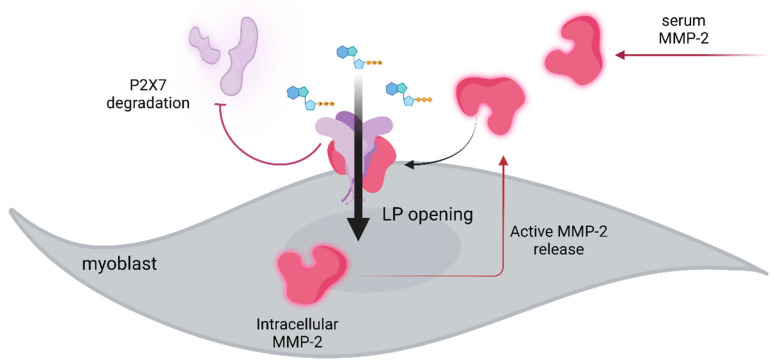
Following P2X7 stimulation with high eATP, the large pore response is activated, leading to active MMP-2 being released by an unknown mechanism. In addition, in organs with physiologically or pathologically permeable capillaries, serum MMP-2 can access P2X7 receptors present on any cell in that tissue. Through direct receptor cleavage, MMP-2 released in a mechanism associated with P2X7 activation or present in the serum abrogates P2X7 functions. However, this inhibitory effect may be further regulated by endogenous MMP-2 inhibitors such as TIMPs, so that the net effect on P2X7 activity can vary even further, BioRender.

## Data Availability

No new data were created or analyzed in this study. Data sharing is not applicable to this article.

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
