# Peer review of "The Role of P2X7 Purinoceptors in the Pathogenesis and Treatment of Muscular Dystrophies"

_ijms, 2023, doi:10.3390/ijms24119434_

Round 1

Reviewer 1 Report

This is a well written and comprehensive review report concerning the role of P2X7 receptor in different muscular dystrophies.

The role played by purinergic signalling and its alteration in skeletal muscle in MD is is well described. Authors highlight how the altered expression of P2X7R in muscle myofibers as well as in myogenic cells may even exacerbate the pathological phenotype and describe two possible explanation to the up regulation of the P2X7R in dystrophic muscles.

Great attention is dedicated to the therapeutic implications of the P2X7R blockage in DMD and sarcoglycanopathies, with compounds under development or already approved for different indications.

 Minor points:

-          When the authors refer to sarcoglycanopathies, the new nomenclature should also be used (i.e. LGMD2D/R3).

-          Pag 3 line 130: a ref concerning the overexpression of P2X7R on dystrophic muscle cells should be provided.

-          Pag 4 lines 175-178: the authors should provide some more information regarding the expression level of other ecto-ATPse and ecto-enzyme in skeletal muscle.

Author Response

Reviewer 1

Minor points:

-          When the authors refer to sarcoglycanopathies, the new nomenclature should also be used (i.e. LGMD2D/R3).

Done

-          Pag 3 line 130: a ref concerning the overexpression of P2X7R on dystrophic muscle cells should be provided.

Done

-          Pag 4 lines 175-178: the authors should provide some more information regarding the expression level of other ecto-ATPse and ecto-enzyme in skeletal muscle.

Relevant information have been included.

Reviewer 2 Report

Comments and suggestons for the authors. 

a)       Which few exceptions (please see below) work is well written and organized and highly detailed. This review manuscript comprehensively cover the main described topics, i.e. The role of purinergic receptor (P2X7) the pathogenesis and treatment of muscular dystrophies. Figures are informative and well rendered. The work present high quality and thus it deserves to be published in IJMS.

b)      As general comment, several notions are well reported but without supporting references. Please include some additional reference

c)       Please include the aim of the work at the end of the introduction. In a similar fashion, the abstract should be rephrased in order to include the aim

d)      For completeness, please include a couple of words on NLRP inflammasome and cancer in the corresponding section. PMID: 33707781

e)      Section 3, lines 133-182, for completeness, these two, recently published detailed reviews on P2X7 should be included PMID: 35267424 and PMID: 34987401

f)        Line 148 “in vitro” should be initalics

g)       The tool used for generating figures should be mentioned in the figures captions

h)      Please check the different font style across section 4 subparagraphs

i)        If present, clinical trials on Muscular dystrophies based on targeting  P2X7 should be mentioned

j)        AZD9056 has also been tested for cancer terapy PMID: 32004973, please include this information

Author Response

Reviewer 2

  1. a)Which few exceptions (please see below) work is well written and organized and highly detailed. This review manuscript comprehensively cover the main described topics, i.e. The role of purinergic receptor (P2X7) the pathogenesis and treatment of muscular dystrophies. Figures are informative and well rendered. The work present high quality and thus it deserves to be published in IJMS.
  2. b)As general comment, several notions are well reported but without supporting references. Please include some additional reference

The manuscript has been revised and additional references included.

  1. c)Please include the aim of the work at the end of the introduction. In a similar fashion, the abstract should be rephrased in order to include the aim

Done

  1. d)For completeness, please include a couple of words on NLRP inflammasome and cancer in the corresponding section. PMID: 33707781

Included

  1. e)Section 3, lines 133-182, for completeness, these two, recently published detailed reviews on P2X7 should be included PMID: 35267424 and PMID: 34987401

Included

  1. f)Line 148 “in vitro” should be initalics

Corrected

  1. g)The tool used for generating figures should be mentioned in the figures captions

Done

  1. h)Please check the different font style across section 4 subparagraphs

Done

  1. i)If present, clinical trials on Muscular dystrophies based on targeting  P2X7 should be mentioned

To our knowledge, no such trials have been initiated.

  1. j)AZD9056 has also been tested for cancer terapy PMID: 32004973, please include this information

Done.